# Structured Graph Learning via Laplacian Spectral Constraints

**Sandeep Kumar**[*]
sandeep0kr@gmail.com

**Jiaxi Ying**[†]
jx.ying@connect.ust.hk

**José Vinícius de M. Cardoso**[†]
jvdmc@connect.ust.hk

**Daniel P. Palomar**[*,†]
palomar@ust.hk

Department of Industrial Engineering and Data Analytics[*]
Department of Electronic and Computer Engineering[†]
The Hong Kong University of Science and Technology, Clear Water Bay, Hong Kong

## Abstract

Learning a graph with a specific structure is essential for interpretability and identification of the relationships among data. It is well known that structured graph learning from observed samples is an NP-hard combinatorial problem. In this paper, we first show that for a set of important graph families it is possible to convert the structural constraints of structure into eigenvalue constraints of the graph Laplacian matrix. Then we introduce a unified graph learning framework, lying at the integration of the spectral properties of the Laplacian matrix with Gaussian graphical modeling that is capable of learning structures of a large class of graph families. The proposed algorithms are provably convergent and practically amenable for large-scale semi-supervised and unsupervised graph based learning tasks. Extensive numerical experiments with both synthetic and real data sets demonstrate the effectiveness of the proposed methods. An R package containing code for all the experimental results is available at https://cran.r-project.org/package=spectralGraphTopology.

## 1 Introduction

Graph models constitute an effective representation of data available across numerous domains in science and engineering [1]. Gaussian graphical modeling (GGM) encodes the conditional dependence relationships among a set of $p$ variables. In this framework, an undirected graph is associated to the variables, where each vertex corresponds to one variable, and an edge is present between two vertices if the corresponding random variables are conditionally dependent [2, 3]. GGM is a tool of increasing importance in a number of fields including finance, biology, statistical learning, and computer vision [4, 5].

For improved interpretability and precise identification of the structure in the data, it is desirable to learn a graph with a specific structure. For example, gene pathways analysis are studied through multi-component graph structures [6, 7], as genes can be grouped into pathways, and connections within a pathway might be more likely than connections between pathway, forming a cluster; a bipartite graph structure yields a more precise model for drug matching and topic modeling in document analysis [8, 9]; a regular graph structure is suited for designing communication efficient deep learning architectures [10, 11]; and a sparse yet connected graph structure is required for graph signal processing applications [12].

Structured graph learning from sample data involves both the estimation of structure (graph connectivity) and parameters (graph weights). While there are a variety of methods for parameter estimation (e.g., maximum likelihood), structure estimation is arguably very challenging due to its combinatorial nature. Structure learning is NP-hard [13, 14] for a general class of graphical models and the effort has been on characterizing families of structures for which learning can be feasible. In this paper, we present one such characterization based on the so-called *spectral properties* of a graph Laplacian matrix. Under this framework, structure learning of a large class of graph families can be expressed as the eigenvalue problem of the graph Laplacian matrix. Our contributions in this paper are threefold. First, we introduce a problem formulation that converts the combinatorial problem of structured graph learning into an optimization problem of graph matrix eigenvalues. Secondly, we discuss various theoretical and practical aspects of the proposed formulation and develop computationally efficient algorithms to solve the problem. Finally, we show the effectiveness of the proposed algorithm with numerous synthetic and real data experiments.

As a byproduct of our investigation, we also reinforce the known connections between graph structure representation and Laplacian quadratic methods (for smooth graph signals) by introducing a procedure that maps a priori information of graph signals to the spectral constraints of the graph Laplacian. This connection enables us to use computationally efficient spectral regularization framework for standard graph smoothing problems to incorporate a priori information.

The rest of the paper is organized as follows. In Section 2, we present related background, the problem formulation, and its connection to smooth graph signal analysis. In Section 3, we first propose a tractable formulation for the proposed problem and then we develop an efficient algorithm and discuss its various theoretical and practical aspects. In Section 4, we show experimental results with real datasets and present additional experiments and the associated convergence proof into the supplementary material. An R package containing the code for all the simulations is made available as open source repository.

## 2 Background and Proposed Formulation

In this section, we review Gaussian graphical models and formulate the problem of structured graph learning via Laplacian spectral constraints.

### 2.1 Gaussian Graphical Models

Let $\mathbf{x} = [x_1, x_2, \ldots, x_p]^T$ be a $p-$dimensional zero mean random vector associated with an undirected graph $\mathcal{G} = (\mathcal{V}, \mathcal{E})$, where $\mathcal{V} = \{1, 2, \ldots, p\}$ is a set of nodes corresponding to the elements of $\mathbf{x}$, and $\mathcal{E} \in \mathcal{V} \times \mathcal{V}$ is the set of edges connecting nodes. The GGM method learns a graph by solving the following optimization problem:

$$\underset{\Theta \in \mathcal{S}_{++}^p}{\text{maximize}} \quad \log \det(\Theta) - \text{tr}(\Theta S) - \alpha h(\Theta), \tag{1}$$

where $\Theta \in \mathbb{R}^{p \times p}$ denotes the desired graph matrix, $\mathcal{S}_{++}^p$ denotes the set of $p \times p$ positive definite matrices, $S \in \mathbb{R}^{p \times p}$ is the sample covariance matrix (SCM) obtained from data, $h(\cdot)$ is the regularization term, and $\alpha > 0$ is the regularization parameter. The optimization in (1) corresponds to the penalized maximum likelihood estimation of the inverse covariance (precision) matrix and also known as Gaussian Markov Random Field (GMRF). With the graph $\mathcal{G}$ inferred from $\Theta$, the random vector $\mathbf{x}$ follows the Markov property, meaning $\Theta_{ij} \neq 0 \iff \{i, j\} \in \mathcal{E} \ \forall \ i \neq j$: implies $x_i$ and $x_j$ are conditionally dependent given the rest [2, 3].

### 2.2 Graph Laplacian

A matrix $\Theta \in \mathbb{R}^{p \times p}$ is called a combinatorial graph Laplacian matrix if it belongs to the following set:

$$\mathcal{S}_{\Theta} = \left\{ \Theta | \Theta_{ij} = \Theta_{ji} \leq 0 \text{ for } i \neq j, \Theta_{ii} = -\sum_{j \neq i} \Theta_{ij} \right\}. \tag{2}$$

The Laplacian matrix $\Theta$ is a symmetric, positive semi definite matrix with zero row sum [15]. The non-zero entries of the matrix encode positive edge weights as $-\Theta_{ij}$ and $\Theta_{ij} = 0$ implies no connectivity between vertices $i$ and $j$. The importance of the graph Laplacian has been well recognized as a tool

for embedding, manifold learning, spectral sparsification, clustering and semi-supervised learning [16, 17, 18, 19, 20, 21, 22]. In addition, structural properties of a large class of important graph families are encoded in the eigenvalues of the graph Laplacian matrix, and utilizing these under the GGM setting is the main goal of the present work.

## 2.3 Structured Gaussian Graphical Models

The goal is to learn matrix $\Theta$ as a Laplacian matrix under some eigenvalue constraints. We introduce a general optimization framework

$$
\begin{aligned}
\underset{\Theta}{\text{maximize}} \quad & \log \text{gdet}(\Theta) - \text{tr}(\Theta S) - \alpha h(\Theta), \\
\text{subject to} \quad & \Theta \in \mathcal{S}_\Theta, \ \boldsymbol{\lambda}(\Theta) \in \mathcal{S}_\lambda,
\end{aligned}
\tag{3}
$$

where $\text{gdet}(\Theta)$ denotes the generalized determinant [23] defined as the product of the non-zero eigenvalues of $\Theta$, $S$ is the SCM (with the mean removed, i.e., $S = \mathbf{xx}^T$) obtained from data $\mathbf{x}$, $\mathcal{S}_\Theta$ is the Laplacian matrix structural constraint (2), $\boldsymbol{\lambda}(\Theta)$ denotes the eigenvalues of $\Theta$, and $\mathcal{S}_\lambda$ is the set containing spectral constraints on the eigenvalues. Precisely $S_\lambda$ will facilitate the process of incorporating the spectral properties required for enforcing structure on the graph to be learned.

From the probabilistic perspective, when the data is generated from a Gaussian distribution $\mathbf{x} \sim \mathcal{N}(\mathbf{0}, \Theta^\dagger)$, then (3) can be viewed as a penalized maximum likelihood estimation of the structured precision matrix of an improper attractive GMRF model [23]. For any arbitrarily distributed data, formulation (3) corresponds to minimizing a penalized log-determinant Bregman divergence problem, and hence this formulation yields a meaningful graph even for distributions that are not GMRFs.

### 2.3.1 Laplacian quadratic and smooth graph signals

In the context of graph signal modeling, the widely used assumption is that the signal/data residing on graphs change smoothly between connected nodes [20, 24, 25, 26]. The trace term in (3) relates to the graph Laplacian quadratic form $\text{tr}(\Theta \mathbf{xx}^T) = \sum_{i,j} -\Theta_{ij}(x_i - x_j)^2$ also known as quadratic energy function, which is used for quantifying smoothness of the graph signals [20]. Smooth graph signal methods are an extremely popular family of approaches for semi-supervised learning. The type of graph used to encode relationships in these learning problems is often a more important decision than the particular algorithm or loss function used, yet this choice is not well-investigated in the literature [24]. Our proposed framework that can learn a graph with a specific structure based on a priori information of the problem at hand is indeed a promising direction for strengthening these approaches.

### 2.3.2 Graph Structure via Laplacian Spectral Constraints

Now, we introduce various choices of $\mathcal{S}_\lambda$ that will enable (3) to learn some important graph structures.

● *$k$-component graph:* A graph is said to be $k-$component connected if its vertex set can be partitioned into $k$ disjoint subsets such that any two nodes belonging to different subsets are not connected. The eigenvalues of any Laplacian matrix can be expressed as:

$$
\mathcal{S}_\lambda = \{\{\lambda_j = 0\}_{j=1}^k, \ c_1 \leq \lambda_{k+1} \leq \cdots \leq \lambda_p \leq c_2\}
\tag{4}
$$

where $k \geq 1$ denotes the number of connected components in the graph, and $c_1, c_2 > 0$ are constants that depend on the number of edges and their weights [15, 19].

● *connected sparse graph:* A sparse graph is simply a graph with not many connections among the nodes. Often, making a graph highly sparse can split the graph into several disconnected components, which many times is undesirable [12, 27]. The existing formulation cannot ensure both sparsity and connectedness, and there always exists a trade-off between the two properties. We can achieve sparsity and connectedness by using the following spectral constraint:

$$
\mathcal{S}_\lambda = \{\lambda_1 = 0, c_1 \leq \lambda_2 \leq \cdots \leq \lambda_p \leq c_2\}
\tag{5}
$$

with a proper choice of $c_1 > 0, c_2 > 0$.

● *$k$-component $d$-regular graph:* All the nodes of a $d$-regular graph have the same weighted degree $d$, i.e., $\sum_{j \in \mathcal{N}_i} -\Theta_{ij} = d, \ \forall \ i = 1, 2, \ldots, p$, where $\mathcal{N}_i$ is the set of neighboring nodes connected to node $i$. This states that the diagonal entries of the matrix $\Theta$ are $d$, $\text{diag}(\Theta) = d\mathbf{1}$. A $k-$component

regular graph structure can be learned by forcing $\mathrm{diag}(\Theta) = d\mathbf{1}$ along with the following spectral constraints

$$\mathcal{S}_\lambda = \{\{\lambda_j = 0\}_{j=1}^k, \ c_1 \leq \lambda_{k+1} \leq \cdots \leq \lambda_p \leq c_2\}, \ \ \mathrm{diag}(\Theta) = d\mathbf{1}. \tag{6}$$

• *cospectral graphs:* In many applications, it is motivated to learn $\Theta$ with specific eigenvalues which is also known as cospectral graph learning [28]. One example is spectral sparsification of graphs [19, 29] which aims to learn a graph $\Theta$ to approximate a given graph $\bar{\Theta}$, while $\Theta$ is sparse and its eigenvalues $\lambda_i$ satisfy $\lambda_i = f(\bar{\lambda}_i)$, where $\{\bar{\lambda}_i\}_{i=1}^p$ are the eigenvalues of the given graph $\bar{\Theta}$ and $f$ is some specific function. Therefore, for cospectral graph learning, we introduce the following constraint

$$\mathcal{S}_\lambda = \{\lambda_i = f(\bar{\lambda}_i), \quad \forall i \in [1, p]\}. \tag{7}$$

## 2.4 Related work and discussion

The complexity of structure learning depends critically on the underlying graph structure and the focus has been on characterizing classes of structures for which learning is feasible. The seminal work [30] established that structure learning for tree-structured graph reduces to a maximum weight spanning tree problem, while the work in [14] presented a characterization based on the local separation property, and proposed a greedy method based on thresholding of sample statistics for learning the following graph structures: Erdos-Renyi random graphs, power law graphs, small world graphs, and other augmented graphs. Sparse graphs have been been widely studied in the high-dimensional setting [31]. A sparse graph under the GGM model (1) is typically learned by introducing an $\ell_1$-norm penalty term, such as Graphical Lasso (GLasso) [32]. But a uniform sparsity is not enough when a specific structure is desired [33, 34]. Recent works extended the GGM to include other structures such as factor models [35], scale-free [36], degree-distribution [37], and overlapping structure with multiple graphical models [34, 38], those methods are restrictive to the particular case and it is difficult to extend them to learn other structures.

A feasible characterization that can enable a $k-$component structured graph learning is still lacking. Existing methods employ a relaxation based approach where they focus on either structure estimation or parameter estimation. The work in [39] can only do structure estimation, while the works in [40, 41, 42] estimate parameters with structure information known already. In recent work, the authors in [6] have developed a two-stage approach for learning a multi-component structure. The method is based on the integration expectation maximization (EM) with the GMM method, which can estimate both the structure and the parameters jointly. However, the costly EM step makes this approach computationally prohibitive for large scale problems.

Finally, several recent publications considered learning different types of graph Laplacians (2) under the GGM setting [26, 43, 44]; however, they do not include spectral constraints and are not able to enforce specific structures onto the graph. Specifically, all these methods are limited to learning a connected graph without structural constraints, or just learn Laplacian weights for a graph with given structure estimates.

### 2.4.1 Discussion

The present work identifies that the spectral characteristics of the graph matrices are a natural and efficient tool for learning structured graphs. The proposed idea is to use the spectral characteristics directly into a graph learning framework. Here, the focus is on utilizing Laplacian spectral constraints under the GGM-type model but the proposed machinery has a much wider appeal. For example, the proposed framework can be easily extended to learn more non-trivial structures (e.g., bipartite and clustered bipartite graph structures) by considering spectral properties of other graph matrices, e.g., adjacency, normalized Laplacian, and signless Laplacian [15, 45, 46, 47]; furthermore, the scope of spectral methods can be easily extended to other important statistical models such as the Ising model [48], Gaussian covariance graphical models [49], Gaussian graphical models with latent variables [50], least-square formulation for graph learning [51], structured linear regression, vector autoregression models [52], and also for the structured graph signal processing applications [21, 53, 54].

# 3 Optimization Method

We reformulate the optimization problem presented in (3) by introducing a graph Laplacian linear operator $\mathcal{L}$ and spectral penalty which, by consequence, transforms the combinatorial Laplacian structural constraints into easier to handle algebraic constraints.

## 3.1 Graph Laplacian operator $\mathcal{L}$

The Laplacian matrix $\Theta$ belonging to $\mathcal{S}_\Theta$ satisfies i) $\Theta_{ij} = \Theta_{ji} \leq 0$, ii) $\Theta \mathbf{1} = \mathbf{0}$, implying the target matrix is symmetric with degrees of freedom of $\Theta$ equal to $p(p-1)/2$. Therefore, we introduce a linear operator $\mathcal{L}$ that transforms a non-negative vector $\mathbf{w} \in \mathbb{R}_+^{p(p-1)/2}$ to the matrix $\mathcal{L}\mathbf{w} \in \mathbb{R}^{p \times p}$ that satisfies the Laplacian constraints ($[\mathcal{L}\mathbf{w}]_{ij} = [\mathcal{L}\mathbf{w}]_{ji} \leq 0$, for $i \neq j$ and $[\mathcal{L}\mathbf{w}] \cdot \mathbf{1} = \mathbf{0}$) as in (2).

**Definition 1.** The linear operator $\mathcal{L} : \mathbf{w} \in \mathbb{R}_+^{p(p-1)/2} \to \mathcal{L}\mathbf{w} \in \mathbb{R}^{p \times p}$ is defined as

$$[\mathcal{L}\mathbf{w}]_{ij} = \begin{cases} -w_{i+d_j} & i > j, \\ [\mathcal{L}\mathbf{w}]_{ji} & i < j, \\ \sum_{i \neq j}[\mathcal{L}\mathbf{w}]_{ij} & i = j, \end{cases}$$

where $d_j = -j + \frac{j-1}{2}(2p - j)$.

We derive the adjoint operator $\mathcal{L}^*$ of $\mathcal{L}$ to satisfy $\langle \mathcal{L}\mathbf{w}, Y \rangle = \langle \mathbf{w}, \mathcal{L}^* Y \rangle$.

**Lemma 1.** The adjoint operator $\mathcal{L}^* : Y \in \mathbb{R}^{p \times p} \mapsto \mathcal{L}^* Y \in \mathbb{R}^{\frac{p(p-1)}{2}}$ is defined by

$$[\mathcal{L}^* Y]_k = y_{i,i} - y_{i,j} - y_{j,i} + y_{j,j},$$

where $i, j \in \mathbb{Z}^+$ satisfy $k = i - j + \frac{j-1}{2}(2p - j)$ and $i > j$.

**Lemma 2.** The operator norm $\|\mathcal{L}\|_2$ is $\sqrt{2p}$, where $\|\mathcal{L}\|_2 = \sup_{\|\mathbf{x}\|=1} \|\mathcal{L}\mathbf{x}\|_F$ with $\mathbf{x} \in \mathbb{R}^{p \times (p-1)/2}$.

*Proof.* Follows from the definitions of $\mathcal{L}$ and $\mathcal{L}^*$: see supplementary material for detailed proof. $\square$

By the definition of the Laplacian operator $\mathcal{L}$ in (1), the set of graph Laplacian constraints in (2) can be expressed as $\mathcal{S}_\Theta = \{\mathcal{L}\mathbf{w} | \mathbf{w} \geq \mathbf{0}\}$, where $\mathbf{w} \geq \mathbf{0}$ means each entry of $\mathbf{w}$ is non-negative. We represent the Laplacian matrix $\Theta \in \mathcal{S}_\Theta$ as $\mathcal{L}\mathbf{w}$.

To ensure sparsity of edges in the learned graph, we use the $\ell_1$-regularization function. Observe that the sign of $\mathcal{L}\mathbf{w}$ is fixed by the constraints $\mathcal{L}\mathbf{w}_{ij} \leq 0$ for $i \neq j$ and $\mathcal{L}\mathbf{w}_{ij} \geq 0$ for $i = j$, the regularization term $\alpha \|\mathcal{L}\mathbf{w}\|_1$ can be written as $\mathrm{tr}(\mathcal{L}\mathbf{w}H)$, where $H = \alpha(2I - \mathbf{1}\mathbf{1}^T)$, which implies $\mathrm{tr}(\Theta S) + \alpha h(\mathcal{L}\mathbf{w}) = \mathrm{tr}(\mathcal{L}\mathbf{w}K)$, where $K = S + H$.

## 3.2 Reformulating problem (3) with graph Laplacian operator

To solve (3), for learning a graph Laplacian $\Theta$ with the desired spectral properties, we propose the following Laplacian spectral constrained optimization problem

$$\underset{\mathbf{w}, \boldsymbol{\lambda}, U}{\mathsf{minimize}} - \log \mathrm{gdet}(U \mathrm{Diag}(\boldsymbol{\lambda}) U^T) + \mathrm{tr}(K \mathcal{L}\mathbf{w}) + \frac{\beta}{2}\|\mathcal{L}\mathbf{w} - U \mathrm{Diag}(\boldsymbol{\lambda}) U^T\|_F^2, \qquad (8)$$

$$\mathsf{subject\ to}\ \ \mathbf{w} \geq 0,\ \boldsymbol{\lambda} \in \mathcal{S}_\lambda,\ U^T U = I.$$

where $\mathcal{L}\mathbf{w}$ is the desired Laplacian matrix which seeks to admit the decomposition $\mathcal{L}\mathbf{w} = U \mathrm{Diag}(\boldsymbol{\lambda}) U^T$, $\mathrm{Diag}(\boldsymbol{\lambda}) \in \mathbb{R}^{p \times p}$ is a diagonal matrix containing $\{\lambda_i\}_{i=1}^p$ on its diagonal, and $U \in \mathbb{R}^{p \times p}$ is a matrix satisfying $U^T U = I$. We incorporate specific spectral properties on $\{\lambda_i\}_{i=1}^p$ by the following spectral penalty term $\frac{\beta}{2}\|\mathcal{L}\mathbf{w} - U \mathrm{Diag}(\boldsymbol{\lambda}) U^T\|_F^2$ with $\mathcal{S}_\lambda$ containing a priori spectral information of the desired graph structure. We introduce the term $\frac{\beta}{2}\|\mathcal{L}\mathbf{w} - U \mathrm{Diag}(\boldsymbol{\lambda}) U^T\|_F^2$ to keep $\mathcal{L}\mathbf{w}$ close to $U \mathrm{Diag}(\boldsymbol{\lambda}) U^T$ instead of exactly solving the constraint. Note that this relaxation can be made tight by choosing sufficiently large or iteratively increasing $\beta$. The penalty term can also be understood as a spectral regularization term, which aims to provide a direct control over the eigenvalues allowing to incorporate additional information via priors. This has been successfully used in matrix factorization applications, see [55, 56, 57, 58, 59] for more details.

We consider solving (8) for learning a $k-$component graph structure utilizing the constraints in (4), where the first $k$ eigenvalues are zero. There are a total of $q = p - k$ non-zero eigenvalues ordered in the given set $\mathcal{S}_\lambda = \{c_1 \leq \lambda_{k+1} \leq \cdots \leq \lambda_p \leq c_2\}$. Collecting the variables in three blocks as $\mathcal{X} = \left( \mathbf{w} \in \mathbb{R}^{p(p-1)/2}, \boldsymbol{\lambda} \in \mathbb{R}^q, U \in \mathbb{R}^{p \times q} \right)$ we develop an algorithm based on the block successive upper-bound minimization (BSUM) framework [60], which updates each block sequentially while keeping the other blocks fixed.

### 3.3 Update of w

At iteration $t + 1$, treating $\mathbf{w}$ as a variable with fixed $\boldsymbol{\lambda}, U$ and ignoring the terms independent of $\mathbf{w}$, we have the following sub-problem:

$$\underset{\mathbf{w} \geq 0}{\text{minimize}} \quad \text{tr}\left( K\mathcal{L}\mathbf{w} \right) + \frac{\beta}{2} \|\mathcal{L}\mathbf{w} - U\text{Diag}(\boldsymbol{\lambda})U^T\|_F^2. \tag{9}$$

The problem (9) is equivalent to the non-negative quadratic program problem

$$\underset{\mathbf{w} \geq 0}{\text{minimize}} \quad f(\mathbf{w}) = \frac{1}{2}\|\mathcal{L}\mathbf{w}\|_F^2 - \mathbf{c}^T\mathbf{w}, \tag{10}$$

which is strictly convex where $\mathbf{c} = \mathcal{L}^*(U\text{Diag}(\boldsymbol{\lambda})(U)^T - \beta^{-1}K)$. It is easy to check that the sub-problem (10) is strictly convex. However, due to the non-negativity constraint ($\mathbf{w} \geq 0$), there is no closed-form solution, and thus we derive a majorization function via the following lemma.

**Lemma 3.** *The function $f(\mathbf{w})$ in (10) is majorized at $\mathbf{w}^t$ by the function*

$$g(\mathbf{w}|\mathbf{w}^t) = f(\mathbf{w}^t) + (\mathbf{w} - \mathbf{w}^t)^T \nabla f(\mathbf{w}^t) + \frac{L}{2}\|\mathbf{w} - \mathbf{w}^t\|^2$$

*where $\mathbf{w}^t$ is the update from previous iteration, $L = \|\mathcal{L}\|_2^2 = 2p$. The condition for the majorization function can be easily checked [61, 62].*

After ignoring the constant terms in Lemma 3, the problem (10) is majorized at $\mathbf{w}^t$ as

$$\underset{\mathbf{w} \geq 0}{\text{minimize}} \quad \frac{1}{2}\mathbf{w}^T\mathbf{w} - \mathbf{a}^T\mathbf{w}, \text{ where } \mathbf{a} = \mathbf{w}^t - \frac{1}{2p}\nabla f(\mathbf{w}^t) \text{ and } \nabla f(\mathbf{w}^t) = \mathcal{L}^*(\mathcal{L}\mathbf{w}^t) - \mathbf{c}. \tag{11}$$

**Lemma 4.** *From the KKT optimality conditions we can obtain the optimal solution as*

$$\mathbf{w}^{t+1} = \left( \mathbf{w}^t - \frac{1}{2p}\nabla f(\mathbf{w}^t) \right)^+, \text{ where } (a)^+ = \max(a, 0). \tag{12}$$

### 3.4 Update for U

At iteration $t + 1$, treating $U$ as a variable, and fixing $\mathbf{w}$ and $\boldsymbol{\lambda}$, we obtain the following subproblem:

$$\underset{U}{\text{maximize}} \quad \text{tr}\left( U^T \mathcal{L}\mathbf{w} U \text{Diag}(\boldsymbol{\lambda}) \right) \quad \text{subject to} \quad U^T U = I_q. \tag{13}$$

**Lemma 5.** *From the KKT optimality conditions the solution to (13) is given by*

$$U^{t+1} = \textit{eigenvectors}(\mathcal{L}\mathbf{w})[k + 1 : p], \tag{14}$$

*that is, the $n - k$ eigenvectors of the matrix $\mathcal{L}\mathbf{w}$ in increasing order of eigenvalue magnitude [63, 64].*

### 3.5 Update for λ

We obtain the following sub-problem for the update of $\boldsymbol{\lambda}$ for given $\mathbf{w}$ and $U$:

$$\underset{\boldsymbol{\lambda} \in \mathcal{S}_\lambda}{\text{minimize}} \quad -\log\det(\text{Diag}(\boldsymbol{\lambda})) + \frac{\beta}{2}\|U^T(\mathcal{L}\mathbf{w})U - \text{Diag}(\boldsymbol{\lambda})\|_F^2. \tag{15}$$

Now, $\boldsymbol{\lambda}$ only contains non-zero eigenvalues in increasing order, we can replace the generalized determinant with the determinant on $\text{Diag}(\boldsymbol{\lambda})$. For notation brevity, we denote the indices for the non-zero eigenvalues $\lambda_i$ from 1 to $q = p - k$ instead of $k + 1$ to $p$. Next the sub-problem (15) can be further written as

$$\underset{c_1 \leq \lambda_1 \leq \cdots \leq \lambda_q \leq c_2}{\text{minimize}} \quad -\sum_{i=1}^{q} \log \lambda_i + \frac{\beta}{2}\|\boldsymbol{\lambda} - \mathbf{d}\|_2^2, \tag{16}$$

where $\boldsymbol{\lambda} = [\lambda_1, \ldots, \lambda_q]^T$ and $\mathbf{d} = [d_1, \ldots, d_q]^T$, with $d_i$ the $i$-th diagonal element of $\mathrm{Diag}(U^T(\mathcal{L}\mathbf{w})U)$. The sub-problem (16) is a convex optimization problem and the solution can be obtained from the KKT optimality conditions. One can solve the convex problem (16) with a solver but not suitable for large scale problems. We derive a tailor-made computationally efficient algorithm, which updates $\boldsymbol{\lambda}$ following an iterative procedure with the maximum number of $q + 1$ iterations. Please refer to the supplementary material for the detailed derivation of the algorithm.

### 3.6  SGL Algorithm Summary

Algorithm 1, which we denote by SGL, summarizes the implementation of the structured graph learning via Laplacian spectral constraints. Note that the eigen-decomposition for the update $U$ is the

---

**Algorithm 1** SGL

    **Input:** SCM **S**, $k, c_1, c_2, \beta$
    $t \leftarrow 0$
    **while** Stopping criteria not met **do**
        Update $\mathbf{w}^{t+1}$ as in (12).
        Update $U^{t+1}$ as in (14).
        Update $\boldsymbol{\lambda}^{t+1}$ as discussed in Section 3.5.
        $t \leftarrow t + 1$
    **end while**
    **Output:** $\hat{\Theta}^{t+1} = \mathcal{L}\mathbf{w}^{t+1}$

---

most demanding task in our algorithm, with a complexity of $O(p^3)$. This is very efficient considering the fact that the total number of parameters to estimate is $O(p^2)$, which also are required to satisfy complex combinatorial-structural constraints. Computationally efficient graph learning algorithms such as GLasso [32] and GGL [26] have similar worst-case complexity, though they learn a graph without any structural constraints. It is implied that the algorithm would be applicable to problems where eigenvalue decomposition can be performed–which nowadays are possible for large scale problems.

**Remark 1.** Apart from learning $k-$component graph, the SGL algorithm can also be easily adapted to learn other graph structures with aforementioned spectral constraints in (5) to (7). Furthermore, SGL can also be utilized to learn classical connected graph structures (e.g., Erdos-Renyi graph, modular graph, grid graph, etc.) just by setting the eigenvalue constraints corresponding to one component graph (i.e., $k = 1$) and $c_1, c_2$ to very small and large values, respectively.

**Theorem 1.** *The sequence $(\mathbf{w}^t, U^t, \boldsymbol{\lambda}^t)$ generated by Algorithm 1 converges to the set of KKT points of* (8).

*Proof:* The detailed proof is deferred to the supplementary material.

## 4  Experiments

In this section, we illustrate the advantages of incorporating spectral information directly into a graph learning framework with real data experiments. We apply SGL to learn similarity graphs from a real categorical animal dataset [65] with binary entries to highlight that it can obtain a meaningful graph for non-Gaussian data as well. We also apply our method to detect biologically meaningful clusters from complex and high-dimensional PANCAN cancer dataset [66]. Performance is evaluated based on visual inspection and by evaluating accuracy (ACC). Additional experiments with different performance measures (e.g., relative error and F-score) for several structures, such as Grid, Modular, and multi-component, noisy multi-component graph structures are shown in the supplementary material..

### 4.1  Animals data set

Herein, animals data set [65, 67] is taken into consideration to learn weighted graphs. The data set consists of binary values (categorical non-Gaussian data) which are the answers to questions such as "is warm-blooded?," "has lungs?", etc. There are a total of 102 such questions, which make up the features for 33 animal categories. Figure 1 shows the results of estimating the graph of the animals

data set using the SGL algorithm, with GGL[1], and GLasso. Graph vertices denote animals, and edge weights representing similarity among them. The input for all the algorithms is the sample covariance matrix plus an identity matrix scaled by $1/3$ (see [26]). The evaluation of the estimated graphs is based on the visual inspection. It is expected that similar animals such as (*ant*, *cockroach*), (*bee*, *butterfly*), and (*trout*, *salmon*) would be grouped together. Based on this premise, it can be seen that the SGL algorithm yields a more clear graph than the ones learned by GGL and GLasso.

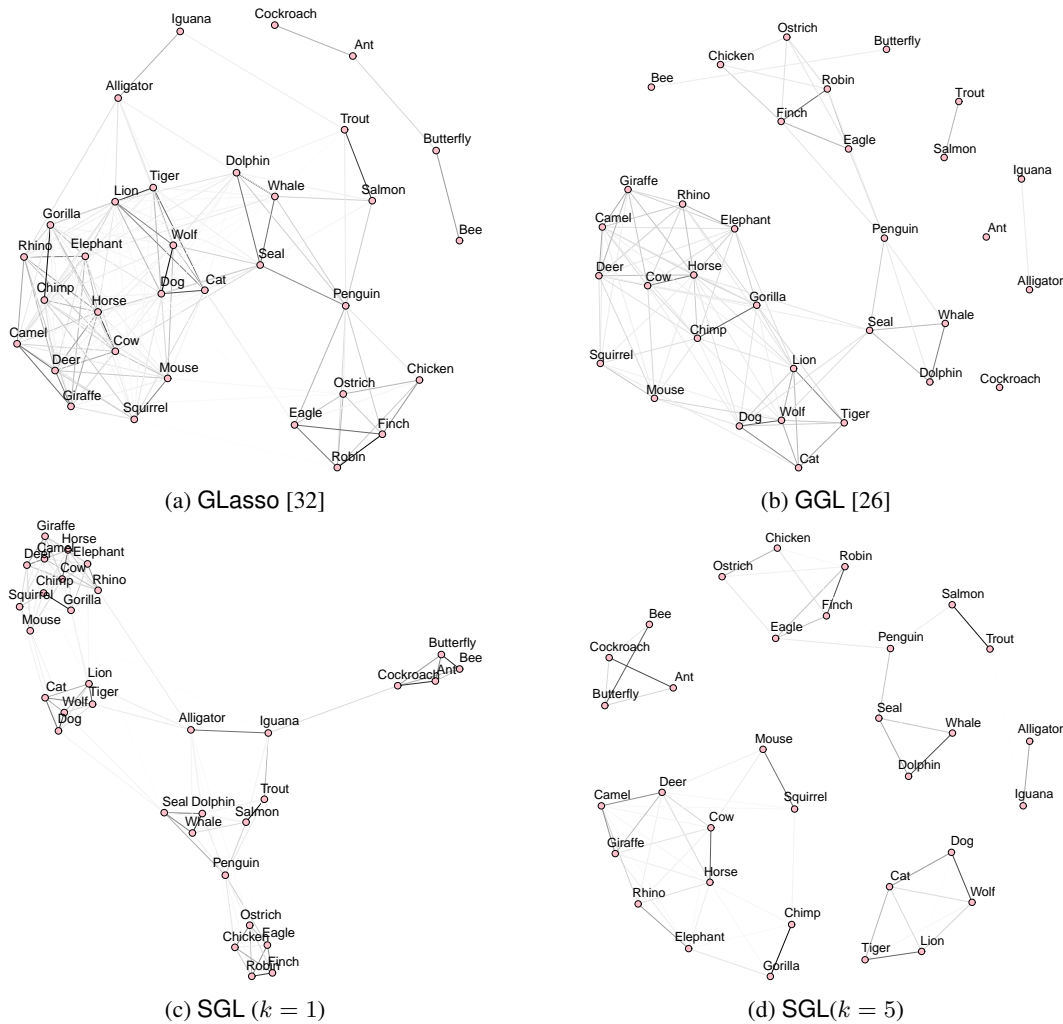

Figure 1: Perceptual graphs of animal connections are obtained by (a) GGL, (b) GLasso, and (c) SGL with $k = 1$, and (d) SGL with $k = 5$. GGL, GLasso split the graph into multiple components due to the sparsity regularization, while SGL with $k = 1$ (connectedness) yields a sparse yet connected graph. (d) SGL with $k = 5$ obtains a graph with 5 components which depicts a more fine-grained representation of animal connection by grouping similar animals in respective components. Furthermore, the animal data is categorical (non-Gaussian) which does not follow the GMRF assumption, the above result also establishes the capability of SGL under mismatch of the data model.

## 4.2 Cancer Genome data set

We consider the RNA-Seq Cancer Genome Atlas Research Network [66] data set available at the UC-Irvine Machine Learning Database [68]. This data set consists of genetic features which map 5 types of cancer namely: breast carcinoma (BRCA), kidney renal clear-cell carcinoma (KIRC), lung

adenocarcinoma (LUAD), colon adenocarcinoma (COAD), and prostate adenocarcinoma (PRAD). In Figure 2, they are labeled with colors *black, blue, red, violet*, and *green*, respectively. The data set consists of 801 labeled samples, in which every sample has 20531 genetic features and the goal is to classify and group the samples, according to their tumor type, on the basis of those genetic features.

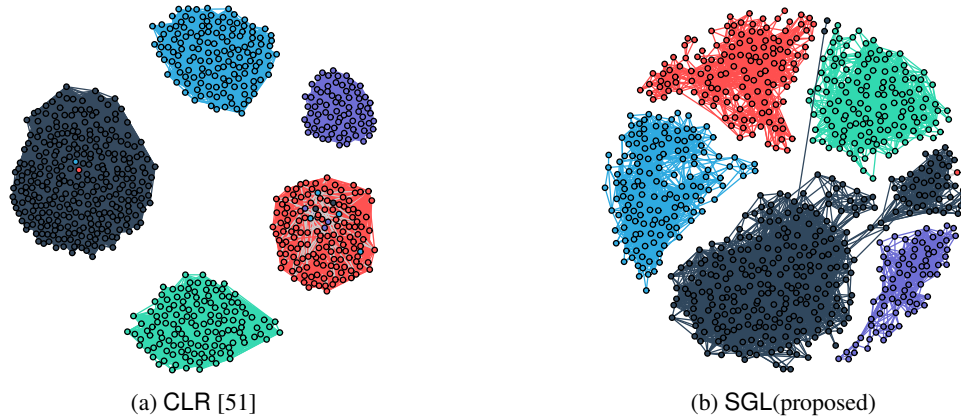

(a) CLR [51]                                          (b) SGL(proposed)

Figure 2: Clustering with (a) CLR method–there are two miss-classified points in the *black* group and 10 miss-classified points in the *red* group, and (b) Clustered graph learned with proposed SGL with $k = 5$ shows a perfect clustering. Furthermore, the graph for the BRCA (black) data sample highlights an inner sub-grouping: suggesting for further biological investigation.

We compare the SGL performance against the state-of-the-art method for graph-based clustering, i.e., constrained Laplacian rank algorithm CLR [51]. CLR uses a well-curated similarity measure as the input to the algorithm, which is obtained by solving a separate optimization problem, while the SGL takes the sample covariance matrix as its input. Still SGL method outperforms CLR, even though the later is a specialized clustering algorithm. The clustering accuracy (ACC)[51] for both the methods are ( CLR=0.9862, SGL=0.99875). The improved performance of the SGL can be attributed to two main reasons i) SGL is able to estimate the graph structure and weight simultaneously, which is essentially an optimal joint procedure, ii) SGL is able to capture the conditional dependencies (i.e., inverse covariance matrix entries) among nodes which consider a global view of relationships, while the CLR encodes the connectivity via the direct pairwise distances. The conditional dependencies relationships are expected to give an improved performance for clustering tasks [6].

To the best of our knowledge, the SGL is the first single stage algorithm that can learn a clustered graph directly from sample covariance matrix data without any additional pre-processing (i.e., learning optimized similarity matrix) or post-processing steps (i.e., thresholding). This makes the SGL highly favorable for large-scale unsupervised learning applications.

## 5   Conclusion

In this paper, we have shown how to convert the combinatorial constraints of structured graph learning into analytical constraints of the graph matrix eigenvalues. We presented the SGL algorithm that can learn structured graphs directly from sample data. Extensive numerical experiments with both synthetic and real datasets demonstrate the effectiveness of the proposed methods. The algorithm enjoys comprehensive theoretical convergence properties along with low computational complexity.

## Acknowledgments

This work was supported by the Hong Kong GRF 16207019 research grant.

## Footnotes

[1]The state-of-the-art algorithm for learning generalized graph Laplacian [26].

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
