[Supplementary Material]

# Supplementary material: Structured Graph Learning Via Laplacian Spectral Constraints

**Sandeep Kumar**[1]
sandeep0kr@gmail.com

**Jiaxi Ying**[2]
jx.ying@connect.ust.hk

**José Vinícius de M. Cardoso**[2]
jvmirca@gmail.com

**Daniel P. Palomar**[1,2]
palomar@ust.hk

Department of Industrial Engineering and Data Analytics[1]
Department of Electronic and Computer Engineering[2]
The Hong Kong University of Science and Technology, Kowloon, Hong Kong

This document contains the proof for Lemma 1 and Theorem 1 along with some additional experiments highlighting the algorithmic performance evaluation and for showing the applicability of the algorithm for learning some popular classical graph structures.

## 1 Proof of Lemma 1

*Proof.* We define an index set $\Omega_t$:

$$\Omega_t := \left\{ l \mid [\mathcal{L}\mathbf{x}]_{tt} = \sum_{l \in \Omega_t} x_l \right\}, \quad t \in [1, p]. \tag{1}$$

For any $\mathbf{x} \in \mathbb{R}^{\frac{p(p-1)}{2}}$, we have

$$\|\mathcal{L}\mathbf{x}\|_F^2 = 2 \sum_{k=1}^{\frac{p(p-1)}{2}} x_k^2 + \sum_{i=1}^{p} ([\mathcal{L}\mathbf{x}]_{ii})^2 \tag{2}$$

$$= 4 \sum_{k=1}^{\frac{p(p-1)}{2}} x_k^2 + \sum_{t=1}^{n} \sum_{i,j \in \Omega_t, \; i \neq j} x_i x_j \tag{3}$$

$$\leq 4 \sum_{k=1}^{\frac{p(p-1)}{2}} x_k^2 + \frac{1}{2} \sum_{t=1}^{n} \sum_{i,j \in \Omega_t, \; i \neq j} x_i^2 + x_j^2 \tag{4}$$

$$= (4 + 2(|\Omega_t| - 1)) \sum_{k=1}^{\frac{p(p-1)}{2}} x_k^2 \tag{5}$$

$$= 2p \|\mathbf{x}\|^2, \tag{6}$$

where the second equality is due to the fact that each $x_k$ only appears twice on the diagonal; the first inequality achieves equality when each $x_k$ is equal; the last equality follows the fact that $|\Omega_t| = p - 1$.

Therefore, by the definition of operator norm, we can obtain

$$\|\mathcal{L}\|_2 = \sup_{\|\mathbf{x}\|=1} \|\mathcal{L}\mathbf{x}\|_F = \sqrt{2p}, \tag{7}$$

concluding the proof.

$\square$

## 2  Derivation for $\lambda'$s update

Next the sub-problem for the $\boldsymbol{\lambda}$ update is

$$\underset{c_1 \leq \lambda_1 \leq \cdots \leq \lambda_q \leq c_2}{\text{minimize}} \quad -\sum_{i=1}^{q} \log \lambda_i + \frac{\beta}{2}\|\boldsymbol{\lambda} - \mathbf{d}\|_2^2, \tag{8}$$

The sub-problem (8) is a convex optimization problem. One can solve the convex problem (8) with a solver (e.g., CVX) but we can do it more efficiently with our algorithm for large scale problems. The solution is obtained by satisfying the KKT optimality conditions.

The Lagrangian of the optimization (8) is

$$L(\boldsymbol{\lambda}, \boldsymbol{\mu}) = -\sum_{i=1}^{q} \log \lambda_i + \frac{\beta}{2}\|\boldsymbol{\lambda} - \mathbf{d}\|_2^2 \tag{9}$$

$$+ \mu_1(c_1 - \lambda_1) + \sum_{i=2}^{q} \mu_i(\lambda_{i-1} - \lambda_i) + \mu_{q+1}(\lambda_q - c_2).$$

The KKT optimality conditions are derived as:

$$-\frac{1}{\lambda_i} + \beta(\lambda_i - d_i) - \mu_i + \mu_{i+1} = 0, \; i = 1, \cdots, q; \tag{10}$$

$$c_1 - \lambda_1 \leq 0; \tag{11}$$

$$\lambda_{i-1} - \lambda_i \leq 0, \; i = 2, \cdots, q; \tag{12}$$

$$\lambda_q - c_2 \leq 0; \tag{13}$$

$$\mu_i \geq 0, \; i = 1, \cdots, q+1; \tag{14}$$

$$\mu_1(c_1 - \lambda_1) = 0; \tag{15}$$

$$\mu_i(\lambda_{i-1} - \lambda_i) = 0, \; i = 2, \cdots, q; \tag{16}$$

$$\mu_{q+1}(\lambda_q - c_2) = 0; \tag{17}$$

Algorithm 1 summarizes the iterative procedure for updating $\boldsymbol{\lambda}$.

**Remark 1.** The problem of the form (8) is popularly known as a regularized isotonic regression problem. The isotonic regression is a well-researched problem that has found applications in numerous domains see [see 1, 2, 3, 4, 5]. To the best of our knowledge, however, there does not exist any computationally efficient method comparable to the Algorithm 1. The proposed algorithm can obtain a globally optimal solution within a maximum of $q+1$ iterations for the $q$-dimensional regularized isotonic regression problem, and can be potentially adapted to solve other isotonic regression problems. The computationally efficient Algorithm 1 also holds an important contribution for the isotonic regression literature.

**Lemma 1.** *The solution of the KKT system* (10)-(17) *is* $\lambda_i = (d_i + \sqrt{d_i^2 + 4/\beta})/2$, *for* $i = 1, \cdots, q$, *if* $c_1 \leq \lambda_1 \leq \cdots \leq c_q \leq c_2$ *hold.*

*Proof.* It is obvious that it conditions $c_1 \leq \lambda_1 \leq \cdots \leq \lambda_q \leq c_2$ hold, then the solutions of the primal and dual variables satisfy all equations. $\square$

We start from the corresponding unconstrained version of the problem (8) whose solution is

$$\lambda_i^{(0)} = \left(d_i + \sqrt{d_i^2 + 4/\beta}\right)/2. \tag{18}$$

---

**Algorithm 1** Update rule for $\lambda_1, \cdots, \lambda_q$

---

1: **Compute:** $\lambda_i = (d_i + \sqrt{d_i^2 + 4/\beta})/2$ for $1 \le i \le q$.
2: **if** $\lambda$ satisfies $c_1 \le \lambda_1 \le \cdots \le \lambda_q \le c_2$ **then**
3:      RETURN $\lambda_1, \cdots, \lambda_q$.
4: **end if**
5: **while** $\lambda$ violates $c_1 \le \lambda_1 \le \cdots \le \lambda_q \le c_2$ **do**
6:      **check situation 1:**
7:      **if** $c_1 \ge \lambda_1 \ge \cdots \ge \lambda_r$ with at least one inequality strict and $r \ge 1$,
8:         Set $\lambda_1 = \cdots = \lambda_r = c_1$.
9:      **end if**
10:     **check situation 2:**
11:     **if** $\lambda_s \ge \cdots \ge \lambda_q \ge c_2$ with at least one equality strict and $s \le q$,
12:        Set $\lambda_s = \cdots = \lambda_q = c_2$.
13:     **end if**
14:     **check situation 3:**
15:     **if** $\lambda_i \ge \cdots \ge \lambda_m$ with at least one equality strict and $1 \le i \le m \le q$,

$$\text{Set } \lambda_i = \cdots = \lambda_m = \left(\bar{d}_{i \to m} + \sqrt{\bar{d}_{i \to m}^2 + 4/\beta}\right)/2, \text{with } \bar{d}_{i \to m} = \frac{1}{m - i + 1} \sum\nolimits_{j=i}^{m} d_j.$$

16:     **end if**
17: **end while**
18: RETURN $\lambda_1, \cdots, \lambda_q$

---

If this solution satisfies all the KKT conditions (10)-(17), then it is also the optimal. Otherwise, each $\lambda_i^{(0)}$ that violates the conditions $c_1 \le \lambda_1^{(0)} \le \cdots \le \lambda_q^{(0)} \le c_2$ needs to be updated.

**Situation 1:** $c_1 \ge \lambda_1^{(0)} \ge \cdots \ge \lambda_r^{(0)}$, implying $c_1 - \frac{1}{c_1 \beta} \ge d_1 \ge \cdots \ge d_r$, where at least one inequality is strict and $r \ge 1$. Without loss of generality, let the $j$-th inequality is strict with $1 \le j \le r$, i.e. $d_j > d_{j+1}$. The KKT optimality conditions for this pare are:

$$-\frac{1}{\lambda_j} + \beta(\lambda_j - d_j) - \mu_j + \mu_{j+1} = 0; \tag{19}$$

$$-\frac{1}{\lambda_{j+1}} + \beta(\lambda_{j+1} - d_{j+1}) - \mu_{j+1} + \mu_{j+2} = 0; \tag{20}$$

$$\lambda_j - \lambda_{j+1} \le 0; \tag{21}$$

$$\mu_i \ge 0, \ i = j, j+1, j+2; \tag{22}$$

$$\mu_{j+1}(\lambda_j - \lambda_{j+1}) = 0; \tag{23}$$

We subtract the first two equations and obtain:

$$2\mu_{j+1} = \mu_{j+2} + \mu_j + \left(\frac{1}{\lambda_j} - \frac{1}{\lambda_{j+1}}\right) + \beta(\lambda_{j+1} - \lambda_j) + \beta(d_j - d_{j+1}) > 0, \tag{24}$$

due to the fact that $d_j > d_{j+1}$ and $\lambda_j \le \lambda_{j+1}$. Since $\mu_{j+1} > 0$, we also have

$$2\mu_j = \mu_{j+1} + \mu_{j-1} + \left(\frac{1}{\lambda_{j-1}} - \frac{1}{\lambda_j}\right) + \beta(\lambda_j - \lambda_{j-1}) + \beta(d_{j-1} - d_j) > 0, \tag{25}$$

where $d_{j-1} \ge d_j$ and $\lambda_{j-1} \le \lambda_j$. Similarly, we can obtain $\mu_j > 0$ with $2 \le j \le r$. In addition,

$$\mu_1 = -\frac{1}{\lambda_1} + \beta(\lambda_1 - d_1) + \mu_2 \tag{26}$$

$$-\frac{1}{c_1} + \beta(c_1 - d_1) + \mu_2 > 0. \tag{27}$$

Totally, we have $\mu_j > 0$ with $1 \le j \le r$. By (15) and (16), we obtain $\lambda_1 = \cdots = \lambda_r = c_1$. Therefore, we update

$$\lambda_1^{(1)} = \cdots = \lambda_r^{(1)} = c_1. \tag{28}$$

**Situation 2:** $\lambda_s^{(0)} \geq \cdots \geq \lambda_q^{(0)} \geq c_2$, implying $d_s \geq \cdots \geq d_q \geq c_2 - \frac{1}{c_2\beta}$, where at least one inequality is strict and $s \leq q$.

Similar to situation 1, we can also obtain $\mu_j > 0$ with $s+1 \leq j \leq m+1$ and thus $\lambda_s = \cdots = \lambda_q = c_2$. Therefore, we update $\lambda_s^{(0)}, \cdots, \lambda_q^{(0)}$ by $\lambda_s^{(1)} = \cdots = \lambda_q^{(1)} = c_2$.

**Situation 3:** $\lambda_i^{(0)} \geq \cdots \geq \lambda_m^{(0)}$, implying $d_i \geq \cdots \geq d_m$, where at least one inequality is strict and $1 \leq i \leq m \leq q$. Here we assume $\lambda_{i-1}^{(0)} < \lambda_i^{(0)}$ ($c_1 < \lambda_1^{(0)}$ if $i = 1$) and $\lambda_m^{(0)} < \lambda_{m+1}^{(0)}$ ($\lambda_q^{(0)} < c_2$ if $m = q$). Otherwise, this will be reduced to situation 1 or 3.

Similar to situation 1, we can also obtain $\mu_j > 0$ with $i+1 \leq j \leq m$ and thus $\lambda_i^{(1)} = \lambda_{i+1}^{(1)} = \cdots = \lambda_m^{(1)}$.

We sum up equations (19) with $i \leq j \leq m$ and obtain

$$-\frac{1}{\lambda_j} + \beta\lambda_j - \frac{1}{m-i+1}(\beta\sum_{j=i}^{m} d_j + \mu_i - \mu_{m+1}) = 0, \quad j = i, \cdots, m. \tag{29}$$

Here we need to use iterative method to find the solution that satisfies KKT conditions. It is easy to check that $\mu_i = \mu_{m+1} = 0$ when $\lambda_{i-1}^{(0)} < \lambda_i^{(0)}$ and $\lambda_m^{(0)} < \lambda_{m+1}^{(0)}$. In that case, according to (29), we have

$$\lambda_j = \left(\bar{d}_{i\to m} + \sqrt{\bar{d}_{i\to m}^2 + 4/\beta}\right)/2, \quad j = i, \cdots, m. \tag{30}$$

where $\bar{d}_{i\to m} = \frac{1}{m-i+1}\sum_{j=i}^{m} d_j$. Therefore, we update $\lambda_i^{(0)}, \cdots, \lambda_m^{(0)}$ by

$$\lambda_i^{(1)} = \cdots = \lambda_m^{(1)} = \left(\bar{d}_{i\to m} + \sqrt{\bar{d}_{i\to m}^2 + 4/\beta}\right)/2. \tag{31}$$

If there exists the case that $\lambda_{i-1}^{(1)} > \lambda_i^{(1)}$, we need to further update $\lambda_{i-1}^{(1)}, \lambda_i^{(1)}, \cdots, \lambda_m^{(1)}$ in the next iteration. It will include two cases to discuss:

1. $\lambda_{i-1}^{(1)}$ has not been updated by (31), implying that $\lambda_{i-1}^{(1)} = \lambda_{i-1}^{(0)} = \left(\bar{d}_{i-1} + \sqrt{d_{i-1}^2 + 4/\beta}\right)/2$. So $\lambda_{i-1}^{(1)} > \lambda_i^{(1)}$ means $d_{i-1} > \bar{d}_{i\to m}$. KKT conditions for this pare are:

$$-\frac{1}{\lambda_{i-1}} + \beta(\lambda_{i-1} - d_{i-1}) - \mu_{i-1} + \mu_i = 0; \tag{32}$$

$$-\frac{1}{\lambda_i} + \beta(\lambda_i - \bar{d}_{i\to m}) - \mu_i + \mu_{m+1} = 0; \tag{33}$$

$$\lambda_{i-1} - \lambda_i \leq 0; \tag{34}$$

$$\mu_p \geq 0, \ p = i-1, i, m+1; \tag{35}$$

$$\mu_i(\lambda_{i-1} - \lambda_i) = 0; \tag{36}$$

We subtract the first two equations and obtain

$$2\mu_i = \mu_{i-1} + \mu_{m+1} + (\frac{1}{\lambda_{i-1}} - \frac{1}{\lambda_i}) + \beta(\lambda_i - \lambda_{i-1}) + \beta(d_{i-1} - \bar{d}_{i\to m}) > 0, \tag{37}$$

and thus $\lambda_{i-1} = \lambda_i = \cdots = \lambda_m$. Then the equation (29) can be written as

$$-\frac{1}{\lambda_j} + \beta\lambda_j - \frac{1}{m-i+2}(\beta\sum_{j=i-1}^{m} d_j + \mu_{i-1} - \mu_{m+1}) = 0, \quad j = i-1, \cdots, m. \tag{38}$$

Hence, we update

$$\lambda_{i-1}^{(2)} = \cdots = \lambda_m^{(2)} = \left(\bar{d}_{(i-1)\to m} + \sqrt{\bar{d}_{(i-1)\to m}^2 + 4/\beta}\right)/2. \tag{39}$$

2. $\lambda_{i-1}^{(1)}$ has been updated by (31), implying that $\lambda_t^{(1)} = \cdots = \lambda_{i-1}^{(1)} = \left(\bar{d}_{t\to(i-1)} + \sqrt{d_{t\to(i-1)}^2 + 4/\beta}\right)/2$ with $t < i-1$. Then $\lambda_{i-1}^{(1)} > \lambda_i^{(1)}$ means $\bar{d}_{t\to(i-1)} > \bar{d}_{i\to m}$.

Similarly, we can also obtain $\lambda_t = \lambda_{t+1} = \cdots = \lambda_m$ by deriving KKT conditions. We sum up equations (10) over $t \leq j \leq m$ and obtain

$$-\frac{1}{\lambda_j} + \beta\lambda_j - \frac{1}{m-t+1}(\beta\sum_{j=t}^{m} d_j + \mu_t - \mu_{m+1}) = 0, \quad j = t, \cdots, m. \tag{40}$$

So we update

$$\lambda_t^{(2)} = \cdots = \lambda_m^{(2)} = \left(\bar{d}_{t\to m} + \sqrt{\bar{d}_{t\to m}^2 + 4/\beta}\right)/2. \tag{41}$$

For the case that $\lambda_m^{(1)} > \lambda_{m+1}^{(1)}$, the update strategy is similar to the case $\lambda_{i-1}^{(1)} > \lambda_i^{(1)}$.

We iteratively check each situation and update the corresponding $\lambda_i$ accordingly. We can check that the algorithm will be terminated with the maximum number of iterations $q+1$ and $c_1 \leq \lambda_1^{(q+1)} \leq \cdots \leq \lambda_q^{(q+1)} \leq c_2$ holds for all variables. Because each updating above is derived by KKT optimality conditions for Problem (8), the iterative-update procedure summarized in Algorithm 1 converges to the KKT point of Problem (8).

## 3 Proof for Theorem 2

*Proof.* The proof of algorithm convergence is partly based on the proof of BSUM in [6]. We first show the linear independence constraint qualification on unitary constraint set $\mathcal{S}_U \triangleq \{U \in \mathbb{R}^{p\times q}|U^TU = I_q\}$.

**Lemma 2.** *Linear independence constraint qualification (LICQ) holds on each $U \in \mathcal{S}_U$.*

*Proof.* We rewrite $\mathcal{S}_U$ as

$$\{U \in \mathbb{R}^{p\times q}|g_{ij}(U) = \sum_{k=1}^{p} u_{ki}u_{kj} - I_{ij}, \forall 1 \leq i \leq j \leq q\}, \tag{42}$$

where $u_{ij}$ and $I_{ij}$ are the elements of $U$ and identity matrix $I$ in $i$-th row and $j$-th column, respectively. It is observed that

$$\nabla g_{ij}(U) = \begin{cases} [0_{p\times(i-1)}; 2u_i; 0_{p\times(q-i)}], & \text{if } i = j; \\ [0_{p\times(i-1)}; u_j; 0_{p\times(j-i-1)}; u_i; 0_{p\times(q-j)}], & \text{otherwise.} \end{cases} \tag{43}$$

We can see $u_i$ from $\nabla g_{ii}(U)$ will only appear in $i$-th column, but $u_i$ from $\nabla g_{ij}(U)$ with $i \neq j$ will not appear in $i$-th column. Consequently, each $\nabla g_{ij}(U)$ cannot be expressed as a linear combination of the others, thus each $\nabla g_{ij}(U)$ is linear independent. $\square$

Now we prove Theorem 2. It is easy to check that the level set $\{(\mathbf{w}, U, \boldsymbol{\lambda})|f(\mathbf{w}, U, \boldsymbol{\lambda}) \leq f(\mathbf{w}^{(0)}, U^{(0)}, \boldsymbol{\lambda}^{(0)})\}$ is compact, where $f(\mathbf{w}, U, \boldsymbol{\lambda})$ is the cost function in Problem (8 in the paper manuscript). Furthermore, the sub-problems ((10) and (16) in the paper manuscript) have unique solutions since they are strictly convex problems and we get the global optima. According to Theorem 2 in [6], we obtain that the sequence $(\mathbf{w}^{(t)}, U^{(t)}, \boldsymbol{\lambda}^{(t)})$ generated by Algorithm 1(i.e., in the paper manuscript) converges to the set of stationary points. Note that $U$ is constrained on the orthogonal Stiefel manifold that is nonconvex, while BSUM framework does not cover nonconvex constraints. But the subsequence convergence can still be established [7] due to the fact that the cost function value here is non-increasing and bounded below in each iteration.

Next we will further show that each limit of the sequence $(\mathbf{w}^{(t)}, U^{(t)}, \boldsymbol{\lambda}^{(t)})$ satisfies KKT conditions of Problem ((8) in the paper manuscript). Let $(\mathbf{w}, U, \boldsymbol{\lambda})$ be a limit point of the generated sequence. The Lagrangian function of (equation (8) in the paper manuscript) is

$$L(\mathbf{w}, U, \boldsymbol{\lambda}, \boldsymbol{\mu}_1, \boldsymbol{\mu}_2, M) = -\log \text{gdet}(\text{Diag}(\boldsymbol{\lambda})) + \text{tr}(K\mathcal{L}\mathbf{w}) + \frac{\beta}{2}\|\mathcal{L}\mathbf{w} - U\text{Diag}(\boldsymbol{\lambda})U^T\|_F^2$$
$$- \boldsymbol{\mu}_1^T\mathbf{w} + \boldsymbol{\mu}_2^T h(\boldsymbol{\lambda}) + \text{tr}\left(M^T(U^TU - I_q)\right), \tag{44}$$

where $\boldsymbol{\mu}_1$, $\boldsymbol{\mu}_2$ and $M$ are dual variables, and $\boldsymbol{\mu}_2^T h(\boldsymbol{\lambda}) = \mu_{2,1}(\alpha_1 - \lambda_1) + \sum_{i=2}^{q} \mu_{2,i}(\lambda_{i-1} - \lambda_i) + \mu_{2,q+1}(\lambda_q - \alpha_2)$ with $\boldsymbol{\mu}_2 = [\mu_{2,1}, \cdots, \mu_{2,q+1}]^T$.

(1) we can see $\boldsymbol{\lambda}$ is derived from KKT conditions of sub-problem (equation (16) in the paper manuscript). Obviously, $\boldsymbol{\lambda}$ also satisfies KKT conditions of Problem (equation (8) in the paper manuscript).

(2) we show $\mathbf{w}$ satisfies KKT conditions (equation (10) in the paper manuscript). The KKT conditions with $\mathbf{w}$ can be derived as:

$$\mathcal{L}^*\mathcal{L}\mathbf{w} - \mathcal{L}^*(U\mathrm{Diag}(\boldsymbol{\lambda})U^T - \beta^{-1}K) - \beta^{-1}\boldsymbol{\mu}_1 = 0; \tag{45}$$

$$\boldsymbol{\mu}_1^T\mathbf{w} = 0; \tag{46}$$

$$\mathbf{w} \geq 0; \tag{47}$$

$$\boldsymbol{\mu}_1 \geq 0; \tag{48}$$

Know $\mathbf{w}$ is derived by KKT system see Lemma (4 in the paper manuscript), we obtain

$$\mathbf{w} - (\mathbf{w} - \frac{1}{L_1}(\mathcal{L}^*\mathcal{L}\mathbf{w} - c)) - \mu = 0, \tag{49}$$

and $c = \mathcal{L}^*(U\mathrm{Diag}(\boldsymbol{\lambda})U^T - \beta^{-1}K)$. So we have

$$\mathcal{L}^*\mathcal{L}\mathbf{w} - \mathcal{L}^*(U\mathrm{Diag}(\boldsymbol{\lambda})U^T - \beta^{-1}K) - \frac{1}{L_1}\mu = 0, \tag{50}$$

Therefore, $\mathbf{w}$ also satisfies KKT conditions (equation (8) in the paper manuscript).

(3) KKT conditions with respect to $U$ are as below:

$$\mathcal{L}\mathbf{w}U\mathrm{Diag}(\boldsymbol{\lambda}) - \frac{1}{2}U(\mathrm{Diag}(\boldsymbol{\lambda})^2 + \beta^{-1}(M + M^T)) = 0; \tag{51}$$

$$U^T U = I_q. \tag{52}$$

Since $U$ admits the first order optimality condition on orthogonal Stiefel manifold, we have

$$\mathcal{L}\mathbf{w}U\mathrm{Diag}(\boldsymbol{\lambda}) - U(U^T\mathcal{L}\mathbf{w}U\mathrm{Diag}(\boldsymbol{\lambda}) - \frac{1}{2}[U^T\mathcal{L}\mathbf{w}U, \mathrm{Diag}(\boldsymbol{\lambda})]) = 0, \tag{53}$$

where $[A, B] = AB - BA$. Note that $U^T\mathcal{L}\mathbf{w}U$ is a diagonal matrix according to the update of $U$. So there must exist a $M$ such that $U$ satisfies (51). Therefore, $(\mathbf{w}, U, \boldsymbol{\lambda})$ satisfies KKT conditions of Problem (equation (8) in the paper manuscript).

$\square$

## 4   Experiments

For synthetic experiments we create several synthetic data sets based on different graph structures $\mathcal{G}$. First, we generate an improper GMRF model parameterized by the true precision matrix $\Theta_{\mathcal{G}}$, which follows the Laplacian constraints in (equation (2) in the paper manuscript) as well as the specific graph structure. Then, a total of $n$ samples $\{\mathbf{x}_i \in \mathbb{R}^p\}_{i=1}^n$ are drawn from the IGMRF model with $\Theta_{\mathcal{G}}$: $\mathbf{x}_i \sim \mathcal{N}(\mathbf{0}, \Theta_{\mathcal{G}}^\dagger)$, $\forall i$. The sample covariance matrix $S$ is computed as,

$$S = \frac{1}{n}\sum_{i=1}^{n}(\mathbf{x}_i - \bar{\mathbf{x}}_i)(\mathbf{x}_i - \bar{\mathbf{x}}_i)^T, \quad \text{with} \quad \bar{\mathbf{x}}_i = \frac{1}{n}\sum_{i=1}^{n}\mathbf{x}_i. \tag{54}$$

Algorithms use the SCM $S$ and prior information regarding target graph families, if available (e.g., number of connected components $k$). We set $c_1$ and $c_2$ to very small and large value, respectively, and the choice of $\beta$ is discussed for each case separately. For each scenario, 20 Monte Carlo simulations

are performed. For performance evaluation, we use following metrics, namely, relative error (RE) and F-score (FS):

$$\text{Relative Error} = \frac{\left\| \hat{\Theta} - \Theta_{\text{true}} \right\|_F}{\left\| \Theta_{\text{true}} \right\|_F}, \quad \text{F-Score} = \frac{2\text{tp}}{2\text{tp} + \text{fp} + \text{fn}}, \tag{55}$$

where $\hat{\Theta} = \mathcal{L}\hat{\mathbf{w}}$ is the final estimation result of the algorithm and $\Theta_{\text{true}}$ is the true reference graph Laplacian matrix, true positive (tp) stands for the case when there is an actual edge and the algorithm detects it; false positive (fp) stands for the case when algorithm detects an edge but no actual edge present; and false negative (fn) stands for the case when algorithm misses an actual edge present. Further, we disregard an edge if its weight value is less than $0.1$. The F-score metric takes values in $[0, 1]$ where 1 indicates perfect structure recovery [8]. To check the performance evolution for each iteration $t$ we evaluate the RE and FS with $\hat{\Theta}^t$. Algorithms are terminated when the relative change in $\mathbf{w}^t$ is relatively small.

## 4.1 Benchmarks

The CGL algorithm proposed in [8] is the state-of-the-art method for estimating a connected combinatorial graph Laplacian matrix from the sample covariance matrix. For synthetic data experiments with connected graph structure (e.g., modular, grid, and connected bipartite), we compare the performance of the SGL algorithm against CGL. Additionally, for more insight, we also compare against some heuristic based approaches. These are i) the pseudo-inverse of the sample covariance matrix $S^\dagger$, denoted as Naive and ii) the solution of following quadratic program $\mathbf{w}_{\text{qp}} = \arg\min_{\mathbf{w} \geq 0} \left\| S^\dagger - \mathcal{L}\mathbf{w} \right\|_F^2$, denoted as QP.

For the comparison on multi-component graph learning, as per our knowledge, there is no existing method to learn graph Laplacian matrix with multiple components (e.g., $k-$component and $k-$component bipartite). Thereby, for the sake of completeness, we compare against Naive and QP, which are expected to give meaningful comparisons for high sample scenarios.

For experiments with real data, we compare the algorithm performance with GLasso [9], GGL[1], constrained Laplacian rank algorithm CLR [10], Spectral clustering [11], and $k-$means clustering. Unlike CGL, the GGL algorithm aims to estimate a generalized graph Laplacian matrix. As observed in [8], GGL performance is always superior than CGL, therefore, for real data we omit the comparison with CGL. Note that the GGL and GLasso cannot estimate the standard Laplacian matrix in (equation (2) in the paper manuscript) , thereby, we cannot compare against those for the synthetic experiments. For CGL, GGL, and GLasso the sparsity parameter $\alpha$ is chosen according to the suggested procedures [8, 12].

## 4.2 Performance evaluation for SGL Algorithm

In this Subsection, we evaluate the performance of the SGL algorithm on grid graph, modular graph, multi-component graph, noisy-multi component graph, and popular synthetic structures for clustering).

### 4.2.1 Grid graph

We consider a grid graph structure denoted as $\mathcal{G}_{\text{grid}}(p)$, where $p = 64$ are the number of nodes, each node is attached to their $4$ nearest neighbors (except the vertices at the boundaries), edge weights are selected randomly uniformly from $[0.1, 3]$. Figure 1 depicts the graph structures learned by SGL and CGL for $n/p = 100$, edges smaller than $0.05$ were discarded. For CGL we use $\alpha = 0.005$ whereas, for SGL, we fix $\beta = 20$ and $\alpha = 0.005$.

Figure 2 compares the performance of the algorithms for different sample size regimes on the grid graph model. This is with respect to the number of data samples, used to calculate sample covariance $S$, per number of vertices (i.e., $n/p$), see (54). For $n/p <= 100$, we fix $\beta = 10$, otherwise we fix $\beta = 100$. Additionally, we fix $\alpha = 0$. For QP and Naive we do not need to set any parameters. It is observed in Figure 2, the SGL algorithm significantly outperforms the baseline approaches: for

(a) True grid graph        (b) CGL        (c) SGL(Proposed)

Figure 1: Sample results of learning $\mathcal{G}_{\mathsf{grid}}(64)$ (a) True grid graph, (b) CGL (RE $= 0.09163$, FS $= 0.8057$), and (c) SGL (RE $= 0.0490$, FS $= 0.9955$).

all the sample ratios SGL can achieve a lower average RE and higher average FS. For instance, to achieve a low RE (e.g., 0.1), SGL requires a lower sample ratio ($n/p = 5$) than Naive ($n/p = 80$), QP ($n/p = 29$) and CGL ($n/p = 30$).

Figure 2: Average performance results for learning Laplacian matrix of a $\mathcal{G}_{\mathsf{grid}}$ graph. The SGL algorithm outperforms Naive, QP, and CGL for all the sample ratios.

#### 4.2.2 Modular graph

We consider a random modular graph, also known as stochastic block model, $\mathcal{G}_{\mathsf{mo}}(p, k, \wp_1, \wp_2)$ with $p = 64$ vertices and $k = 4$ modules (subgraphs), where $\wp_1 = 0.01$ and $\wp_2 = 0.3$ are the probabilities of having an edge across modules and within modules, respectively. Edge weights are selected randomly uniformly from $[0.1, 3]$. Figure 3 illustrates the graph learning performances under different nodes to sample ratio ($n/p$). It is observed in Figure 2, the SGL and the CGL algorithm significantly outperforms the Naive and QP. Furthermore, for low sample ratio (i.e., $n/p < 2$) SGL achieves better performance than CGL, while they perform similarly for a higher sample ratio (i.e., $n/p > 2$).

#### 4.2.3 Multi-component graph

We consider to learn a multi-component graph also known as block-structured graph denoted as $\mathcal{G}_{\mathsf{mc}}(p, k, \wp)$, with $p = 64$, $k = 4$ and $\wp = 0.5$, where $p$ is the number of nodes, $k$ is the number of components, and $\wp$ is the probability of having an edge between any two nodes inside a component while the probability of having an edge between any two nodes from different components is zero. Edge weights are selected randomly uniformly from $[0.1, 3]$. Figure 4 illustrates the graph learning performances of different methods in terms of average RE and FS.

#### 4.2.4 Effect of the parameter $\beta$

In the current subsection, we study the effect of the parameter $\beta$ on the algorithm performance in terms of RE and FS. It is observed from Figure 5 that a large $\beta$ enables the SGL to obtain a low RE

Figure 3: Average performance results for learning Laplacian matrix of a modular graph $\mathcal{G}_{\text{mo}}$ with four modules. The proposed SGL for $\beta = 100, \alpha = 0$ method outperforms the base line approaches.

Figure 4: Average performance results as, a function of the number of samples, for learning Laplacian matrix of a 4-component graph. The SGL method demonstrates good performance for multi-component graph learning, and significantly outperforms the baseline approaches Naive and QP.

and a high FS. For a large $\beta$, the formulation puts more weight on the relaxation term so as to make it closer to the spectral constraints. In addition, along with the increase of $\beta$ the RE and FS tend to be stable. But empirically it is observed that a huge $\beta$ slows down the convergence speed of the algorithm.

Figure 5: Effect of the parameter $\beta$ on the SGL algorithm. We consider here estimating of a multi-component graph structure $\mathcal{G}_{\text{mc}}(32, 4, 0.5)$ edge weights drawn randomly uniformly from [0,1]. It is observed from that a large $\beta$ enables the SGL to obtain a low RE and high FS.

### 4.2.5   Multi-component graph: noisy setting

Here we aim to learn a multi-component graph under noisy setting. At first we generate a 4 component graph $\mathcal{G}_{\text{mc}}(20, 4, 1)$ with equal number of nodes in all the components, the nodes inside a component are fully connected and the edges are drawn randomly uniformly from $[0, 1]$. Then we add random

noise to all the in-component and out component edges. The noise is an Erdos-Renyi graph $\mathcal{G}_{\text{ER}}(p, \wp)$, where $p = 20$ is the number of nodes, $\wp = 0.35$ is the probability of having an edge between any two pair of nodes, and edge weights are randomly uniformly drawn from $[0, \kappa]$. Specifically, we consider a scenario where each sample $\mathbf{x}_i \sim \mathcal{N}(\mathbf{0}, \Theta_{\text{noisy}}^{\dagger})$ used for calculating SCM as in (54) is drawn from the noisy precision matrix,

$$\Theta_{\text{noisy}} = \Theta_{\text{true}} + \Theta_{\text{ER}}, \tag{56}$$

where $\Theta_{\text{true}}$ is the true Laplacian matrix and $\Theta_{\text{ER}}$ is the noise Laplacian matrix, which follows the ER graph structure. Figure 6 illustrates an instance of the SGL performance for noisy-multi component graph with fixed $n/p = 30$, $\beta = 400$, $\alpha = 0.1$, and $\kappa = 0.45$.

(a) $\Theta_{\text{true}}$          (b) $\Theta_{\text{noisy}}$          (c) $\Theta_{\text{learned}}$

(d) $\mathcal{G}_{\text{true}}$          (e) $\mathcal{G}_{\text{noisy}}$          (f) $\mathcal{G}_{\text{learned}}$

Figure 6: An example of estimating a 4-component graph. Heat maps of the graph matrices: (a) the ground truth graph Laplacian matrix $\Theta_{\text{true}}$, (b) $\Theta_{\text{noisy}}$ after being corrupted by noise, (c) $\Theta_{\text{learned}}$ the learned graph Laplacian with a performance of $(\text{RE}, \text{FS}) = (0.210, 1)$, which means a perfect structure recovery even in a noisy setting that heavily suppresses the ground truth weights. The panels (d), (e), and (f) correspond to the graphs represented by the Laplacian matrices in (a), (b), and (c), respectively.

### 4.2.6 Multi-component graph: components number mismatch

For learning a multi-component graph structure, SGL requires the knowledge of the number of components $k$, as a prior information, which is a common assumption for similar frameworks. If not available, one can infer $k$ by using existing methods for model selection e.g., cross validation, Bayesian information criteria (BIC), or Akaike information criteria (AIC). Furthermore, we also investigate the performance when accurate information about the true number of clusters is not available.

We consider an experiment involving model mismatch: the underlying Laplacian matrix that generates the data has $j$ number of components but we actually use $k$, $k \neq j$, number of components to estimate it. We generate a $k = 7$ multi-component graph $\mathcal{G}_{\text{mc}}(49, 7, 1)$, the edge weights are randomly uniformly are drawn from $[0, 1]$. Additionally, we consider a noisy model as in (56) i.e.,

$\Theta_{\text{noisy}} = \Theta_{\text{true}} + \Theta_{\text{ER}}$, where the noise is an Erdos-Renyi graph $\mathcal{G}_{\text{ER}}(49, 0.25)$ with edge randomly uniformly drawn from $[0, 0.45]$. Figure 7 shows an example where the underlying graph has seven components, and we apply the SGL algorithm with $j = 2$. As we can see, even though the number of components is mismatched and the data is noisy, the SGL algorithm is still able to identify the true structure with a reasonable performance in terms of F-score and average relative error. The take away from Figure 7 is that, even in the lack of true information regarding the number of components in a graph, the graph learned from the SGL algorithm can yield an initial approximate graph very close to the true graph, which can be used as an input to other algorithms for post-processing to infer more accurate graph.

(a) $\Theta_{\text{true}}$ with $k = 7$       (b) $\Theta_{\text{noisy}}$       (c) $\Theta_{\text{learned}}$ with $k = 2$

(d) $\mathcal{G}_{\text{true}}$ with $k = 7$       (e) $\mathcal{G}_{\text{noisy}}$       (f) $\mathcal{G}_{\text{learned}}$ with $k = 2$

Figure 7: Heat maps of the graph matrices: (a) the ground truth graph Laplacian of a seven-component graph $\Theta_{\text{true}}$, (b) $\Theta_{\text{noisy}}$ after being corrupted by noise, (c) $\Theta_{\text{learned}}$ the learned graph Laplacian with a performance of $(\text{RE}, \text{FS}) = (0.18, 0.81)$. The panels (d), (e), and (f) correspond to the graphs represented by the Laplacian matrices in (a), (b), and (c), respectively. For Figure 7 (c) and (f) we are essentially getting results corresponding to a two-component graph, which is imperative from the usage of spectral constraints of $k = 2$. It is observed that the true graph (d) with $k = 7$ components are contained exactly in the learned graph (f), the extra edges, which are due to the inaccurate spectral information when removed from 7 (f) can yield the true graph. One can use some simple post-processing techniques (e.g., thresholding of elements in the learned matrix $\Theta$), to recover the true component structure.

Figure 8 depicts the average performance of SGL as a function of $k$. The settings for the experiment is same as in Figure 7, except now we use different number of components information for each instances. It is observed that the SGL has its best performance when $k$ matches with the true number of the components in the graph. This also suggests that the SGL algorithm has the potential to be seamlessly integrated with model selection techniques to dynamically determine the number of clusters to use, in a single algorithm [13, 14, 15].

### 4.2.7 Popular multi-component structures

Here we consider the classical problem of clustering for some popular synthetic structures. To do that, we generate 100 nodes per cluster distributed according to structures colloquially known as *two*

Figure 8: Average performance results as a function of the number of components $k$: best results are obtained for true number of components. As we can note, the performance is monotonically increasing and eventually reaches a perfect F-score when $k = 7$.

(a) two moons  (b) two circles  (c) spirals

(d) three circles  (e) worms  (f) helix 3d

Figure 9: SGL is able to perfectly classify the data points according to the cluster membership for all the structures.

*moons*, *two circles*, *three spirals*, *three circles*, *worms* and *helix 3d*. Figure 9 depicts the results of learning the clusters structures using the proposed algorithm SGL.

## 4.3   Different Spectral Constraint

As a concrete example, we reported a detailed analysis of the most used case of $k$-components (equation (4) in the manuscript) and single component graph eq (equation (5) in the manuscript). Now we implement the 4th case (equation (7) in the paper manuscript) in Figure 10 presented below.

(a) Ground truth      (b) Constraint eq (7)      (c) Constraint eq (5)      (d) Eigenvalues

Figure 10: Experiment for a connected graph with $p = 15$ nodes and $n/p = 10$. (a) True graph; (b) Graph learned with exact spectral constraint (RE= 0.19, FS=0.97); (c) Graph learned with only connected graph spectral constraint (RE= 0.34, FS=0.87). This demonstrates that more spectral information helps improve the graph estimation results.

## 4.4 Clustering of **animal** data set

Figure 11 compares the clustering performance of the SGL method for $k = 10$ clusters against the state-of-the-art clustering algorithms: (a) $k-$means clustering, (b) spectral clustering [2], (c) CLR, and (d) SGL with $k = 10$. It is remarked that all the algorithms except the SGL are designed only for clustering (grouping) task and are not capable of specifying further connectivity inside a group while SGL is capable of doing both the task of clustering and connectivity(edge weights) estimation jointly.

(a) $k-$means

(b) Spectral [11]

(c) CLR [10]

(d) SGL with $k = 10$(proposed)

Figure 11: All the methods obtain 10 components intending to group similar animals together. Clustering with $k-$means and spectral methods yield components with un-common(possibly wrong) groupings. For example, in (a) *seal, cow, horse* are grouped together: characteristics of seal does not seem to fit with the cow and horse, and in (b) *cockroach, lion, iguana, tiger, ant, alligator* are grouped together which is also in contrary to the expectation. On the other hand, it is observed that both CLR (c) and SGL (d) are able to obtain groupings of animal adhering to our general expectation of the animal behaviors. Although both the results vary slightly, the final results from both the methods are meaningful. For example, CLR groups all the insects (*bee, butterfly, cockroach, ant*) together in one group while SGL splits them into two groups, one with *ant, cockroach* and another with *bee, butterfly*. On the other hand, SGL groups the herbivore mammals (*horse, elephant, giraffe, deer, camel, rhino, cow*) together in one group, while CLR splits these animals into two groups, one containing *rhino, elephant* and another group containing the rest.

## Footnotes

[1]Code for the methods CGL, GGL is available at https://github.com/STAC-USC/Graph_Learning

[2] Code for spectral and $k-$means is available at https://cran.r-project.org/web/packages/kernlab [16]