[Reviews · NeurIPS 2019]

Reviewer 1



The authors present a new formulation to structured graph learning related to Gaussian graphical modelling. The paper contains an interesting theoretical section and is well illustrated. However, I also have a few remaining questions and suggestions: - It seems, if I understand it correctly, that not a true underlying structure is assumed but one can rather play with different hypotheses as a user. For example in section 2.4 and section 3.6 remark 1 the authors mention different possible scenarios related to Erdos-Renyi graph, grid graph, scale free networks etc. I feel that in this sense the term structured graph learning is potentially somewhat misleading. - A main result is eq (8) which reformulates (3) with graph Laplacian operator. At this point I am somewhat puzzled whether this should be interpreted rather as a new numerical implementation framework or as an important novel theoretical result? - Though it is interesting that the authors comment on convexity issues of sub-problems in section 3.3, it is not clear whether an alternative convex formulation to (8) would exist or not. Another commonly used regularization technique, which is not mentioned is nuclear norm regularization, which often leads to convex formulations. It would be good if the authors could comment on nuclear norm regularization. - On p.5 line 186 the authors mention that || L w ||_1 = tr(L w H). Is this a new result? If yes, I suggest to explicitly formulate it as a Lemma; otherwise, please provide a reference to the literature. - is the proposed method also applicable to large scale problems? if yes, how? I have read the authors' response.

Reviewer 2



This paper proposes a new algorithm for graph structure learning, which has broad applicability to a number of fields. The proposed method seems sound, with potential applicability to non-trivially structured data. At a glance, the chief claims, lemmas, and convergence seem correct. It would be nice if Algorithm 1 and/or Figure 9 from the supplementary material fit in the main body of the paper, but it is understandable if it does not. Overall, this is a solid paper and I have no real criticisms or concerns - my only wish is that the full supplement could fit in the main body of the paper, which is clearly outside the authors' control! The supplement really adds a lot more context to the main paper. More experiments in the main body of the paper would always be nice, but given the focus of the paper, and the rigorous description of the algorithm and its properties, it is understandable why the experimental section is relatively limited or relegated to the appendix - given the R code is directly included and will be released, it should be straightforward for practitioners to apply this method to their domain of choice. Given the importance of other related methods in practice, I think this paper has potential for high impact in a number of domains. POST REBUTTAL: The rebuttal is clear, and addresses concerns brought up by one of the other reviewers, while also adding further context and nuance to the work. Thank you to the authors for such clear explanation. My score remains unchanged largely because my original comments still stand - I like this paper, the method proposed seems very useful, and the code release is a huge plus.

Reviewer 3



Learning graphs from data is an important problem finding many practical applications. This paper contributes by a new regularization strategy that employs spectral constraints of graph Laplacian. The resulting algorithm appears to be novel and technically sound. However, I think this paper will benefit significantly from major revision making the practical relevance of the proposed approach clearer: 1) The authors originally suggested four different spectral constraints but only one of them (k-component graph) was actually evaluated. Implementing and evaluating more than one constraints in this framework could help understand the nature of this strategy. 2) The resulting algorithm has been evaluated on synthetic data sets and two real-world data sets. Evaluation on real-world data sets has been done only based on `visual inspection’. Neither objective criteria (e.g., the BIC used in [14]) nor user study aggregating subjective user opinions have been employed for real-world data sets. Two-dimensional visualizations of clustering results on two simple data sets does not seem to effectively demonstrate the usefulness of the proposed algorithm. Minimally, the authors should provide an intuitive discussion on what practical relevance can `k-component graph’ have for graph learning. To summarize, the current reviewer is not sure if the results on these simple data sets reflect the actual performance of the proposed algorithm on challenging real world problems. Minor comments - The authors should discuss how `k’ should be determined in practical applications. - The proposed framework does not actually enforce the spectral constraints. Instead, the authors replace the hard constraints with soft spectral regularization term that simply measures the squared Frobenius norm from a matrix (in a spectral decomposition form) that satisfies the constraints (the third term in Eq. 8). Quantifying the violation of the original spectral constraints in this scenario could help. Alternatively, the possibility of `having sufficiently large beta’ could be substantiated in terms of the optimization trajectory. - After rebuttal: Thanks for the rebuttal. I increased my score to 6. My main concerns were 1) implementing and evaluating only the k-component graph case among other constraints that the authors suggested; and 2) assessing the performance only based on visual inspection. I think the rebuttal addressed some of my concerns. Suggestions for improvement: 1) I would suggest including the results corresponding to Eq. 6 and Eq. 7, convincingly demonstrating the `general' utility of the proposed spectral regularization framework or alternatively, remaining focused on the k-component graph (including the 1-component graph) case and adjusting the title and main claims accordingly. 2) The experiments could be significantly strengthened by evaluating objectively the resulting algorithm in the context of real-world applications that the authors mentioned in section 3 of the rebuttal, e.g., semi-supervised classification.

[Author Response · NeurIPS 2019]

# Authors Feedback Paper ID: 6220

We thank the reviewers for their valuable comments and for acknowledging the novelty of our work. We hope to address adequately the concerns raised by the reviewers.

1) *"Visual inspection"* For the cancer data set, we agree that we should have included a numerical result. We apologize for overlooking this. Here are the values for clustering accuracy (ACC)[50], (SGL=0.99875, CLR=0.9862). For the animal data set, visualization of the animal connections is a standard practice to evaluate a graph learning algorithm [26, 63, 65], and the performance can be judged based on the intuition that similar animals should be strongly connected.

2) *"Different Spectral Constraints"* As a concrete example, we reported a detailed analysis of the most used case of $k$-components eq (4) and single component graph eq (5). Following reviewer's suggestion, we implement the 4th case eq (7) in Figure 1 presented below.

|          (a) Ground truth          |          (b) Constraint eq (7)          |          (c) Constraint eq (5)          |          (d) Eigenvalues          |

Figure 1: Experiment for a connected graph with $p = 15$ nodes and $n/p = 10$. (a) True graph; (b) Graph learned with exact spectral constraint (RE= 0.19, FS=0.97); (c) Graph learned with only connected graph spectral constraint (RE= 0.34, FS=0.87). This demonstrates that more spectral information helps improve the graph estimation results.

3) *"What practical relevance can a $k$-component graph have?"* $k$-component graph is a widely studied graph structure, found in a variety of applications, e.g., spectral graph theory, clustering, classification, community detection, and pattern matching [6, 7, 18, 39, 41, 42, 44, 45, 50]. $k-$component SGL is an unsupervised learning technique for clustering data and learning the connections. Such a method can be used for simultaneous clustering and graph learning: a much-needed tool for a variety of applications including gene classification and their pathways analysis [6,7].

4) *"term structured graph learning is potentially somewhat misleading"* The framework requires a prior knowledge about the underlying graph structure to be estimated. Erdos Renyi and Grid graphs are estimated under connected constraint eq(5).

5) *"Other points"* A new formulation eq(8) with graph Laplacian operator is a new numerical implementation, which simplifies the design of the algorithm. The sub problem in eq(9) is convex with respect to $\mathbf{w}$, but it is difficult to find a convex formulation for eq(8), due to the eigenvalue constraint. We proposed a graph learning framework viable to several graph structures by considering different eigenvalue constraints. Nuclear norm regularization in our situation could only apply to $k$-component graph learning, while our current formulation is more flexible. $\alpha||\mathcal{L}\mathbf{w}||_1 = \mathrm{tr}(\mathcal{L}\mathbf{w}\mathbf{H})$ is a useful identity which is proposed in [26].

The major computational complexity of our algorithm is the eigenvalue decomposition. Thus our algorithm would be applicable to problems where eigenvalue decomposition can be performed–which nowadays are possible for large scale problems. The parameter '$k$' could be determined by some prior knowledge according to specific applications, e.g., via model selection. More interestingly, it is observed in Fig. 7(in the supplementary file) that the result is still acceptable when '$k$' is inaccurate. It is observed in $k$-component graph learning experiments that the estimated graph always has exactly $k$ components for both synthetic and real-world data, though the original problem is relaxed. In addition, we can see in Fig. 5 that the algorithm achieves a good performance in terms of Average RE by setting a sufficiently large $\beta$, and the performance remains stable for larger $\beta$ but with a decrease in convergence speed.



[Meta-Review · NeurIPS 2019]

The submission proposes a new approach to structured graph learning related to Gaussian graphical modelling and eigenvalue problems. The reviewers were unanimous in their opinion that the paper should be accepted at NeurIPS, and were appreciative of the interesting and broadly applicable method, the theoretical foundations contained within the supplementary material, and the accompanying code release.